# Comparative Analyses Reveal the Genetic Mechanism of Ambergris Production in the Sperm Whale Based on the Chromosome-Level Genome

**DOI:** 10.3390/ani13030361

**Published:** 2023-01-20

**Authors:** Chuang Zhou, Kexin Peng, Yi Liu, Rusong Zhang, Xiaofeng Zheng, Bisong Yue, Chao Du, Yongjie Wu

**Affiliations:** 1Key Laboratory of Bioresources and Ecoenvironment (Ministry of Education), College of Life Sciences, Sichuan University, Chengdu 610064, China; 2Key Laboratory of Sichuan Province for Fishes Conservation and Utilization in the Upper Reaches of the Yangtze River, Neijiang Normal University, Neijiang 641000, China; 3Baotou Teachers College, Baotou 014060, China

**Keywords:** sperm whale, ambergris, chromosome-level genome, positive selection, gene family expansion, *LIPE*, cytochrome P450

## Abstract

**Simple Summary:**

The sperm whale (*Physeter macrocephalus*) is famous for the production of ambergris, while the underlying mechanism remains little known. In this study, comparative genomics analyses based on the chromosome-level genome of the sperm whale were performed. The expanded gene families and positive selected genes (PSGs) were found to be functionally enriched in biological pathways related to steroids, terpenoids, aldosterone, etc. Meanwhile, two predicted damaging missense mutations were found in the PSG LIPE that were important in the lipid and cholesterol metabolism. Furthermore, the HSD and CYP genes were mapped to the chromosome of the sperm whale, and phylogenetic analysis of CYP genes found relatively expanded subfamilies related to steroid and xenobiotics metabolism. Hence, these results may shed light on the genetic mechanism of ambergris production in sperm whales.

**Abstract:**

Sperm whales are a marine mammal famous for the aromatic substance, the ambergris, produced from its colon. Little is known about the biological processes of ambergris production, and this study aims to investigate the genetic mechanism of ambergris production in the sperm whale based on its chromosome-level genome. Comparative genomics analyses found 1207 expanded gene families and 321 positive selected genes (PSGs) in the sperm whale, and functional enrichment analyses suggested revelatory pathways and terms related to the metabolism of steroids, terpenoids, and aldosterone, as well as microbiota interaction and immune network in the intestine. Furthermore, two sperm-whale-specific missense mutations (Tyr393His and Leu567Val) were detected in the PSG LIPE, which has been reported to play vital roles in lipid and cholesterol metabolism. In total, 46 CYP genes and 22 HSD genes were annotated, and then mapped to sperm whale chromosomes. Furthermore, phylogenetic analysis of CYP genes in six mammals found that CYP2E1, CYP51A and CYP8 subfamilies exhibited relative expansion in the sperm whale. Our results could help understand the genetic mechanism of ambergris production, and further reveal the convergent evolution pattern among animals that produce similar odorants.

## 1. Introduction

The sperm whale (*Physeter macrocephalus*) is the sole extant species of the genus *Physeter*. As the largest toothed whale, adult male sperm whales can measure up to 20.7 m long and weigh up to 80 tons [1]. The sperm whale is a cosmopolitan species ranging through all deep oceans of the world, its food varies a lot between different regions and seasons, while squid are always the most abundant food items [2,3,4]. The sperm whale was listed in The IUCN Red List of Threatened Species as Vulnerable A1d in 2008 in light of the threats of biological resource use and pollution [5]. For centuries, sperm whales were hunted for commercial purposes, and one of the most cherished products is ambergris, a coprolith from the digestive system of sperm whales [1]. Ambergris has been prized for over a thousand years as medicine, condiment, aphrodisiac, or perfume [6]. It is a solid, waxy, flammable substance of a dull grey or blackish color with a special scent [7], often collected as jetsam on beaches or found in the carcasses of dead sperm whales [8]. Using DNA sequencing technology, the fact that the biological origin of jetsam ambergris is the sperm whale was proved directly [9]. The chief chemical composition of ambergris is ambrein and variable proportions of co-occurring steroids and terpenoids, especially coprosterol, epicoprosterol, pristane, and cholesterol [8,10]. Ambrein is a tricyclic triterpene alcohol that could be the main contributor to the special odor of ambergris [11,12]. In addition, as its name implies, coprosterol is well known to occur in animal feces, and both coprosterol and epicoprosterol are steroids that are presumably formed from cholesterol [13]. Nowadays, the use of gas chromatography–mass spectrometry (GC–MS) and nuclear magnetic resonance spectroscopy (NMR) makes it easy to analyze the identities and quantities of substances occurring in ambergris [10,14], while the circumstances that induce the production of ambergris are poorly understood and have seldom been validated.

For centuries, scientists developed various theories to illustrate the origin and process of ambergris production. The biliary theory proposed that ambergris might be a biliary concentration, considering the similarity between the ambrein and cholesterol [8]. The fecal theory held the view that ambergris was raised from transformations of substances contained in normal feces. It was thought that the formation of ambergris began from an indigestible mass formed in the stomach. The mass was mainly composed of squid beaks, the principal food of sperm whales. If this mass blocked the intestine, the intestinal wall would react by absorbing water from the feces-impregnated mass, thus causing it to become more solid. With the continuation of this process, the mass would increase in size through the accretion of additional solid layers, before finally being excreted to the outside [8,15]. The fecal theory was held to be true for a long time, until Lambertsen developed the pathology theory [16], which received more approval. The pathology theory suggests that ambergris is a pathological substance arising from irritation of the lining of the gut caused by the horny beaks of squid passing undigested with the feces. Strong evidence for this theory includes the fact that the probability of the occurrence of ambergris is as low as one in every hundred sperm whales, without sex bias, with most ambergris being found in the intestine or caecum of sick whales. It was considered that ill whales form a substance known as ‘calculus’, which would either be excreted, or remain inside, eventually causing the death of the whale [17].

In recent years, studies on ambergris have mostly focused on the biochemical processes likely to happen in the gut during the production of ambergris in the sperm whale. A comparative study showed that there was a significant difference in δ^13^C relative isotopic composition between ambrein and co-occurring sterols from the jetsam ambergris samples, which suggested that ambrein originated via a different biosynthetic mechanism from that of other sterols, and it can be further hypothesized that the formation of ambrein might occur via bacterial production of bicyclic polypodenols in vivo [12]. Furthermore, the metagenome analysis of sperm whale found some pathogenic microbes in the gut, and the results further showed the gut microbiota had a coevolutionary relationship with its host [18]. However, to date, there is little evidence illustrating the genetic mechanisms of ambergris production in the sperm whale. Meanwhile, the rapid development of sequencing technology has made available the genome of the sperm whale in high quality, thus providing a novel opportunity to perform evolutionary adaptation research [19,20].

In this study, we conducted comparative genomics analyses among 11 mammals to identify the contributing factor to the biochemical process of ambergris production, aiming to reveal the genetic mechanisms of ambergris production in the sperm whale, and further to provide insights into the potential convergent evolution pattern of animal-resourced odorant formation. 

## 2. Materials and Methods

### 2.1. Genome Data Collection

The whole genomes of sperm whale, mouse, human, dog, horse, greater horseshoe bat, Arabian camel, pig, central Asian red deer, cattle, and African bush elephant were downloaded from NCBI (https://www.ncbi.nlm.nih.gov/, accessed on 14 March 2022 ) (the accession numbers were sperm whale, GCF_002837175.2; common bottlenose dolphin, GCF_011762595.1; mouse, GCF_000001635.27; human, GCF_000001405.40; dog, GCF_014441545.1; horse, GCF_002863925.1; greater horseshoe bat, GCF_004115265.1; Arabian camel, GCF_000803125.2; pig, GCF_000003025.6; central Asian red deer, GCA_010411085.1; cattle, GCA_021234555.1; African bush elephant, GCF_000001905.1).

### 2.2. Gene Family and Phylogenetic Tree Construction

To identify the gene family of the 11 species, we put the genome sequences in OrthoFinder with default parameters [21]. Mafft [22] was used to find one-to-one orthologous genes with default parameters. Using PAL2NAL [23], the corresponding CDS alignments were back-translated from the corresponding protein alignments. The aligned blocks were removed by Gblocks [24]; then, the CDS alignments that remained in each family were concatenated to form one supergene for each species. Afterwards, we used RAxML under the GTRGAMMA substitution model [25] to construct the phylogenetic tree. African bush elephant was set as the outgroup. Divergence times between species were inferred by MCMCtree in the PAML package [26]. We used three calibration points retrieved from the TimeTree database (http://www.timetree.org/, accessed on 14 March 2022) to calibrate our phylogeny including data for African bush elephant and human (99–109 million years ago), human and dog (91–101 million years ago), dog and cattle (76–82 million years ago), and pig and cattle (59–66 million years ago).

### 2.3. Gene Family Expansion and Contraction

To figure out the evolutionary dynamics of genes, we performed the expansion and contraction of the orthologous gene in 12 species (sperm whale, common bottlenose dolphin, mouse, human, dog, horse, greater horseshoe bat, Arabian camel, pig, central Asian red deer, cattle and African bush elephant) with CAFE [27] based on the phylogenetic tree. The random birth and death process model and a *p*-value of 0.05 was used to detect significantly expanded or contracted gene families. Subsequently, KEGG and GO enrichment analyses of expanded gene families were performed using KOBAS [28].

### 2.4. Positive Selection Analysis

To identify potential positively selected genes (PSGs) in the sperm whale, we used Mafft [22] to align the one-to-one orthologous genes retrieved from the results in OrthoFinder of 11 species. The conserved CDS alignments were extracted by Gblocks [24], and the CDS alignments remained of each one-to-one orthologous gene were used for later identification of PSGs. With the sperm whale set as the foreground branch and other species as background branches, the branch-site model of CODEML in PAML 4.7 [26] was used to detect the potential PSGs. According to the null hypothesis, the Ka/Ks ratio for all codons in all branches must be ≤1. While according to the alternative hypothesis set, the foreground branch included codons evolving at Ka/Ks > 1. Then, a maximum likelihood ratio test was performed to compare the two models. Genes under positive selection were affirmed with a corrected *p*-value (<0.05) and contained at least one positively selected site with a posterior probability > 0.99 according to a Bayes Empirical Bayes analysis [29]. 

### 2.5. Functional Enrichment Analysis

The functional enrichment analysis of the identified PSGs was performed via Metascape (http://metascape.org, accessed on 17 March 2022) [30,31] using the default parameters. Pathway and enrichment analyses were carried out by selecting the genomics sources: KEGG Pathway, GO Biological Processes, GO Cellular Components, and GO Molecular Functions. THe 20 enriched terms with the lowest *p*-value were selected and presented. 

### 2.6. Identification and Chromosomal Distribution of CYPs and HSDs

Using the homology prediction method, the known cytochrome P450 (CYP) and hydroxysteroid dehydrogenases (HSD) sequences of humans were compared to the genome sequences of the other five species to obtain the target genes. Then, the reference CYP and HSD protein sequences were used to blast the six studied mammalian genomes by using TBLASTN (E-value = 1 × 10^−5^) [32]. The longest match with the lowest E-value was selected for subsequent analysis. If the fragment gene sequences belonged to the same query protein, Solar [33] would be used to merge them. Moreover, the gene location of functional genes were plotted on chromosomes by TBtools [34] to better visualize the CYP and HSD gene organizations.

### 2.7. Phylogenetic Analysis of CYPs

The amino acid sequences of functional CYP genes obtained from six species (sperm whale, cattle, central Asian red deer, pig, Arabian camel, and human)were aligned and compared using the MAFFT 7 [35] software package. The gaps were removed using trimAl [36]. Subsequently, we used IQ-TREE [37] to construct a rootless phylogenetic tree based on the best-fit model (JTT + R10) using the ML method for 1000 rounds of bootstrapping algorithm. At the same time, CD-HIT [38] was used to cluster the protein sequences of functional CYP genes. The phylogenetic tree showed the degree of similarity based on the homology of each sequence and the functional CYP genes identified in these six species were divided into families and subfamilies.

## 3. Results

### 3.1. Phylogenetic Analysis and Divergence Time Estimation

Comparing 12 mammalian genomes, 24,904 orthogroups were detected in the sperm whale. Meanwhile, 5028 orthogroups possessed one-to-one orthologous genes. Subsequently, a phylogenetic tree was constructed based on 12 mammalian genomes. The phylogenetic tree indicated that sperm whale and the common bottlenose dolphin were within a subclade that likely derived from a common ancestor ~ 36.6 Ma. The series relationships within Artiodactyla were recovered as (Camelidae + (Suidae + ((Delphinidae + Physeteridae) + (Cervinae + Bovidae)))) (Figure 1).

### 3.2. Gene Family Expansion Related to Ambergris Production

The expansion and contraction of gene families is thought to be important in adaptive phenotypic diversification [39]. We detected the expansion and contraction of gene families in the sperm whale genome, and then used them to perform functional distribution analysis. In total, 1207 and 1090 gene families were found to be expanded and contracted in the sperm whale, respectively. Functional enrichment analysis identified expanded gene families (ADCY9, ATF1, ATF4, ATP1A1, ATP1A2, ATP1A3, ATP1A4, ATP2B2, ATP2B3, CALM1, CALM3, CALML3, GNAS, and ITPR3) related to aldosterone synthesis and secretion (map04925), and the expanded gene families are highlighted blue in Figure 2. The expansion of these gene families might be adaptive to the production of ambergris. Furthermore, we found TRBC1 to be annotated in the intestinal immune network for IgA production (map04672), and GSTK1 was found in the metabolism of xenobiotics by cytochrome P450 (map00980), which might account for the pathological formation of ambergris and the physiological adaptation of sperm whales.

### 3.3. Positive Selection and Functional Enrichment

Positive selection analysis at the whole genome level can provide insights into the genetic basis of differences between species, as well as the dynamics of genome evolution. To identify potential positive selection genes in the sperm whale, we analyzed 5028 one-to-one orthologous genes, and found 321 of them to be under strong positive selection in the sperm whale. KEGG and GO enrichment analyses were conducted on the basis of PSGs in the sperm whale, and the 20 categories with the lowest *p*-values are presented as a bar graph in Figure 3. 

Specifically, KEGG distribution analysis found three pathways and their corresponding PSGs that might be involved in ambergris production (Table 1). LIPE, ADCY2, ATF6B, and CACNA1I were annotated in aldosterone synthesis and secretion (map04925) (Figure 2). LSS and SLC9A3R2 were detected in the steroid biosynthesis (map001001) and adosterone-regulated sodium reabsorption (map049601) pathways, respectively. 

### 3.4. Sperm Whale-Specific Missense Mutations

After identifying the PSGs and the functional enrichment of PSGs in sperm whales, we found one PSG LIPE in the aldosterone synthesis and secretion (map049254) pathway that contained two mutations that were sperm whale specific when considered in comparison to other mammals whose genomes were available. The two mutations in LIPE were Tyr393His and Leu567Val (Figure 4), and both missense mutations were predicted to be damaging according to PolyPhen 2 [40] (Table 2). Furthermore, we assessed damaging effects on the protein structure of Tyr393His and Leu567Val in LIPE (Figure 5). The LIPE gene coded the production of Hormone-sensitive Lipase (HSL) in human chromosome 19. It is a neutral cholesterol ester hydrolase that plays an essential role in mediating the hydrolysis of diacylglycerol and triacylglycerol [41,42]. The activities of HSL and mRNA have been found in a variety of tissues [42]. In particular, they play an important role in intestinal cholesterol homeostasis, participating in acylglycerol hydrolysis in jejunal enterocytes and cholesteryl ester hydrolysis throughout the small intestine [42,43,44]. In view of the vital roles of LIPE in the lipid and cholesterol metabolic process, the missense mutations in LIPE are likely to have a significant influence on the production of ambergris in sperm whales.

### 3.5. CYPs and HSDs

Cytochrome P450s (CYPs) and hydroxysteroid dehydrogenases (HSDs) are two important classes of enzymes that participate in the origin and evolution of vertebrate steroids (Figure 6) [45]. CYPs are monooxygenases that play important roles in the oxidative, peroxidative and reductive metabolism of numerous and diverse endogenous compounds such as steroids, bile acids, and fatty acids as well as various man-made chemicals, including environmental chemicals, pollutants and drugs [46,47]. HSDs basically belong to two distinct protein phylogenies and function in the biosynthesis and inactivation of steroid hormones in different tissues [48,49].

By plotting the functional CYP and HSD genes on the sperm whale chromosome, we detected a total of 46 CYP genes and 22 HSD genes mapped to the sperm whale chromosomes (Figure 7). Chromosome 20 was the most abundant in CYP genes (*n =* 7), followed closely by chromosome 11 (*n =* 6), chromosome 2 (*n =* 5) and chromosome 14 (*n =* 5). Other chromosomes had no more than four CYP genes. With respect to HSD genes, chromosome 6 had the most (*n =* 4), while other chromosomes had fewer than two HSDs. Furthermore, neither CYP genes nor HSD genes were detected on chromosomes 1, 3, 13, and 19.

To figure out the draft pattern of CYP gene family of the sperm whale, we constructed an ML tree using the CYP genes from six species, including sperm whales, cattle, central Asian red deer, pigs, Arabian camels, and humans. In total, we detected 18 CYP families encompassing 52 putative genes from all six mammals, and there were 50 putative genes in 18 CYP families in sperm whales (Figure 8), which is less than the 57 functional genes reported in humans [50]. In particular, the CYP2E1, CYP51A, and CYP8 subfamilies were found to be relatively expanded in sperm whales compared with the other five mammals, elucidates the adaption and functional processes surrounding ambergris production. 

## 4. Discussion

### 4.1. Steroid and Terpenoid Metabolism in Sperm Whale

Only a few animal-originated materials are used in perfumery. The four with the most fame are ambergris (sperm whale), musk (musk deer), civet (civet cats) and castoreum (beaver) [51]. Despite there being some special substances in these odorants, such as ambrein in ambergris and muscone in the musk, these odorants still have a shared composition, which includes steroids, terpenoids and lipid acid etc. [8,10,52]. Therefore, understanding the metabolism mechanism of steroids and terpenoids in the sperm whale can not only help us understand the production of ambergris, but also provide insight into the convergent evolutionary patterns among these odorant-producing animals. In this study, we found PSGs such as LIPE and LSS to be functionally distributed in pathways or terms related to ambergris production. The LSS gene (also called OSC) encodes the protein lanosterol synthase, which is implicated as an initial four-ringed sterol intermediate in cholesterol biosynthesis [53]. It has been reported that the mutation of the LSS gene can cause cholesterol-deficiency-associated cataracts [54], kidney injury [55] and alopecia [56], suggesting the importance of LSS in steroid metabolism homeostasis. It is worth mentioning that we found two sperm-whale-specific missense mutations in the PSG LIPE. LIPE is the gene that codes hormone-sensitive lipase (HSL), not only playing essential roles in lipolysis, but also hydrolyzing tri-, di- and monoacylglycerols, as well as cholesterol [42,57,58].

This evidence for the genetic adaption of steroid and terpenoid metabolism in the sperm whale echoes previous studies performed on other odorant-producing animals. Take the musk deer, for instance: genome analysis showed a considerable number of genes involved in the musk metabolism pathways to be related to steroid and terpenoid biosynthesis; meanwhile, aldosterone-regulated sodium reabsorption was found to play a vital role in musk secretion [59]. Furthermore, transcriptome analyses of the musk gland of wild Siberian musk deer (Moschus moschiferus) found several candidate differentially expressed genes (DEGs) including UGT1A4, SULT2B1, CYP2B6, which are highly involved in pathways and terms related to the biological processes of steroids, steroid hormones, terpenoids, cholesterol, fatty acids, etc. [60]. In conclusion, our results not only illustrate the mechanism of ambergris production based on steroids and terpenoids metabolism, but shed further light on convergent evolutionary patterns among odorant-producing animals.

### 4.2. Aldosterone Synthesis and Secretion in Sperm Whale

According to the fecal theory, ambergris arises from the transformation of substances contained in the normal feces, especially indigestible food such as horny beaks, of sperm whales [8]. Aldosterone is an essential steroid hormone synthesized from cholesterol, primally under the catalysis of the cytochrome P450 superfamily, playing key roles in the regulation of systemic blood pressure through the absorption of sodium and water [61,62]. Understanding the synthesis and secretion of aldosterone can provide insight into the functional adaptation of sodium and water homeostasis in the sperm whale, and thus can probide further understanding of the formation of ambergris. 

In our results, KEGG distribution analysis found three pathways related to steroids or aldosterone, and their corresponding expanded genes and PSGs were detected. In the aldosterone synthesis and secretion (map04925) pathway, fourteen expanded gene families (ADCY9, ATF1, ATF4, ATP1A1, ATP1A2, ATP1A3, ATP1A4, ATP2B2, ATP2B3, CALM1, CALM3, CALML3, GNAS, and ITPR3) were found, and four PSGs (ADCY2, ATF6B, LIPE, CACNA1I) were annotated. Interestingly, for both ADCY9 and ATF4 in the expanded gene families, PSGs with close relatedness were found: ADCY2 and ATF6B, respectively. ADCY2 and ADCY9 are different isoforms of adenylyl cyclases (AC), playing similar functions in many cellular processes as the key enzymes in mammalian heterotrimeric GTP-binding protein (G protein)-dependent signal transduction [63,64]. Moreover, ATP1A encodes Na^+^/K^+^ ATPase [65] and ATP2B encodes Ca^2+^-ATPase [66] on the plasma membrane, both of which are responsible for the ion homeostasis of cells. It is worth mentioning that we found two sperm-whale-specific missense mutations in PSG LIPE. LIPE is the gene that codes hormone-sensitive lipase (HSL), which not only plays an essential role in lipolysis, but also hydrolyzes tri-, di- and monoacylglycerols, as well as cholesterol [42,57,58]. In another KEGG pathway, aldosterone-regulated sodium reabsorption (map049601), one PSG SLC9A3R2 was annotated. SLC9A3R2 encodes a member of the NHERF family of PDZ scaffolding proteins, which mediate many cellular processes, including plying roles in intestinal sodium absorption [67,68].

### 4.3. Adaptation of Commensal Gut Microbiota and Intestinal Immune Network 

There is little doubt that the formation of ambergris occurs in the intestine of sperm whales. According to the pathology theory, ambergris is a pathological substance arising from irritation of the lining of the gut [8]. However, the entire biosynthesis process remains unclear. A comparative study showed a significant difference in δ^13^C relative isotopic composition between ambrein and co-occurring sterols from jetsam ambergris samples of sperm whales, which suggests that ambrein originated via a different biosynthetic mechanism from that of other sterols, and it can be further hypothesized that the formation of ambrein might occur via bacterial production of bicyclic polypodenols in vivo [12]. Meanwhile, the commensal gut microbiota, especially the pathogenic microbes of the sperm whale, have been reported, with the results showing that the gut microbiota of sperm whale has a coevolutionary relationship with its host [18]. 

In this study, we found five expanded gene families (CTNNB1, RAC1, CAV3, DOCK1, and ACTR3) to be functionally annotated in the bacterial invasion of epithelial cells pathway (map05100). These genes play roles in the interaction with invasive bacteria on the membranes of epithelial cells. In addition, TRBC1 was annotated in the intestinal immune network for IgA production (map04672). This gene codes a constant region of the T-cell receptor beta chain and participates in antigen-binding and immunoglobulin-receptor-binding activities [69]. Our results illustrate the potential mechanism of interaction between the gut microbiota and the host as well as immune adaptation in the digestive system in the sperm whale, which may contribute to bettering our understanding of ambergris production.

### 4.4. Evolution and Adaptation of CYP

Cytochrome P450 (CYP) are enzymes containing a heme protoporphyrin bound to cell membranes. The multigene superfamily of CYP encodes more than 50,000 enzymes that can be distributed across various tissues [70]. Mammalian CYP can perform the function of the metabolism and bioactivation of endogenous and exogenous compounds, and the substrates of these actions include steroids, prostaglandins, fatty acids, drugs, chemical carcinogens, and xenobiotics [71,72]. In this study, we found that the CYP2E1, CYP51A, and CYP8 families exhibited a relative expansion in sperm whales compared to in the other five mammals.

Firstly, both the CYP51 and CYP8 families play crucial roles in cholesterol homeostasis. CYP51A is a subfamily in the sterol 14α-demethylation CYP51 family that catalyzes a rate-limiting reaction following the cyclization of squalene to form lanosterol or cycloartenol in sterol biosynthesis, finally intermediating the formation of cholesterol in animals [73,74,75]. In our results, the CYP8 family exhibited a relative expansion compared to the other five species, and the CYP7A1 of the sperm whale was clustered in the CYP8 family. To some extent, CYP7A is considered to be functionally related to CYP8B, with both of them being key enzymes in the bile acid and sterol biosynthetic pathway, catalyzing the oxygenation of sterols from an alpha surface in the middle of the steroid skeleton [76]. CYP7A1 is the first and rate-limiting enzyme of the bile acid biosynthetic pathway, producing two primary bile acids, cholic acid (CA) and chenodeoxycholic acid (CDCA), and CYP8B1 subsequently catalyzes the synthesis of CA and regulates bile acid synthesis in the liver as well as cholesterol absorption in the intestine [77,78,79]. 

Furthermore, the relative expansion of CYP2E1 sheds light on the adaptation of xenobiotics metabolism in sperm whales. CYP2E1 is a major catalyst for the oxidation of various small-molecule chemicals that are suspected to be cancer promoting, such as benzene, ethylene dichloride, trichloroethylene, and so on [71,80]. Interestingly, we also found GSTK1 to be expanded in the metabolism of xenobiotics by cytochrome P450 (map00980). In the human digestive system, the expression of CYP2E1 and GSTK1 was found to be higher in both the gastric and colon tumor tissues [81,82]. As the top animals in the food chain in the ocean, marine mammals are very susceptible to bioaccumulation and biomagnification [83]. Several studies have shown that the effects of xenobiotic exposure on marine mammals can be studied by measuring their CYP profile, including harp seals (Phoca groenlandica), fin whales (*Balaenoptera physalus*), striped dolphins (*Stenella coeruleoalba*), northern bottlenose whales (*Hyperoodon ampullatus*), and polar bear (*Ursus maritimus*) [84,85,86]. Therefore, the expansion of CYP2E1 and GSTK1 provided consultation for the further study of the adaptation of xenobiotics metabolism in sperm whales. It is reasonable to believe that more and more CYP and corresponding enzymes will be detected in marine mammals with the development of technology, helping to promote our understanding and the conservation of these creatures.

## 5. Conclusions

In summary, we used comparative genome analysis to reveal the genetic mechanism of ambergris production in sperm whales at the chromosome level. In total, 1207 expanded gene families and 321 PSGs were annotated and functionally enriched in pathways related to steroid metabolism and aldosterone synthesis, as well as intestinal microbiota interaction and the immune network. Two sperm-whale-specific missense mutations (Tyr393His and Leu567Val) were detected in LIPE, which might play crucial roles in lipid and cholesterol metabolism. The chromosome distribution pattern of CYP genes and HSD genes was shown. Thereafter, phylogenetic analysis of CYP genes found CYP2E1, CYP51A and CYP8 subfamilies to exhibit a relative expansion in sperm whales, revealing the adaptation of steroid and xenobiotics metabolism. This study contributes to our understanding of the genetic mechanism of ambergris production in sperm whales, and further reveals the convergent evolution patterns among animals that produce odorants.

## Figures and Tables

**Figure 1 animals-13-00361-f001:**
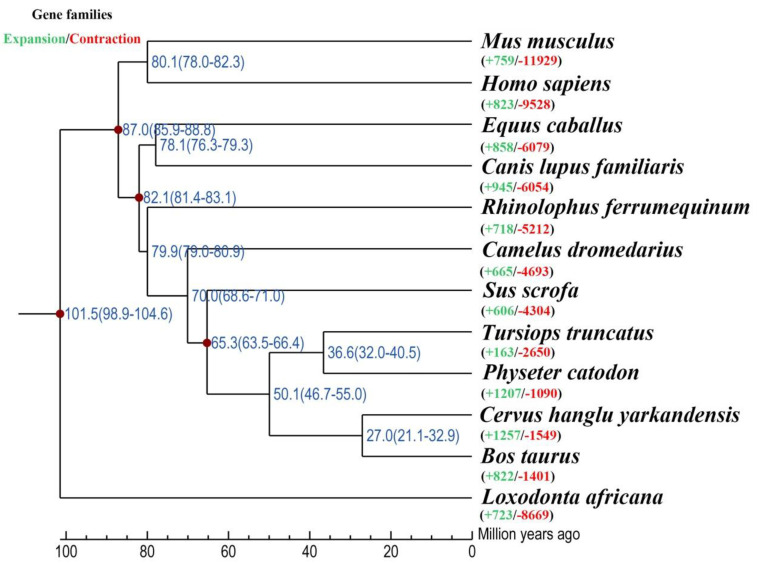
Phylogenetic tree constructed using one-to-one orthologous gene of the 12 mammalian species studied.

**Figure 2 animals-13-00361-f002:**
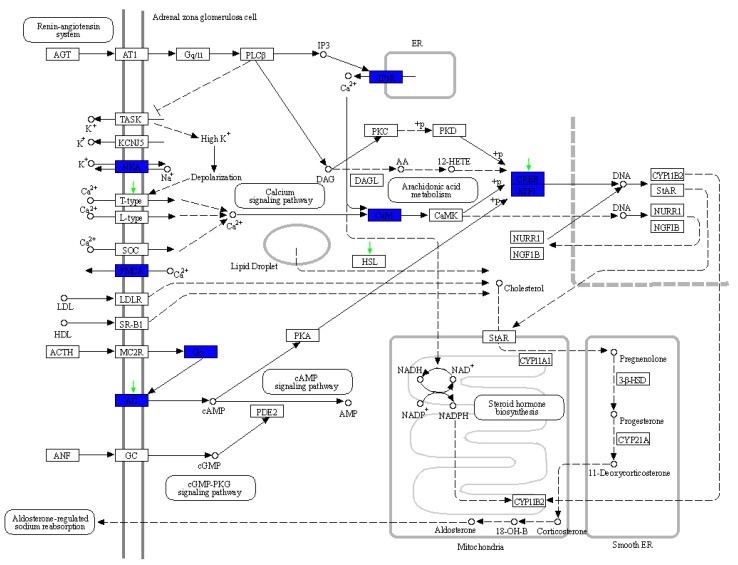
The expansion gene families and positively selected genes (PSGs) annotated in the aldosterone synthesis and secretion (map04925) pathway. The expansion gene and PSGs found in this study are indicated by blue rectangles and green arrows, respectively.

**Figure 3 animals-13-00361-f003:**
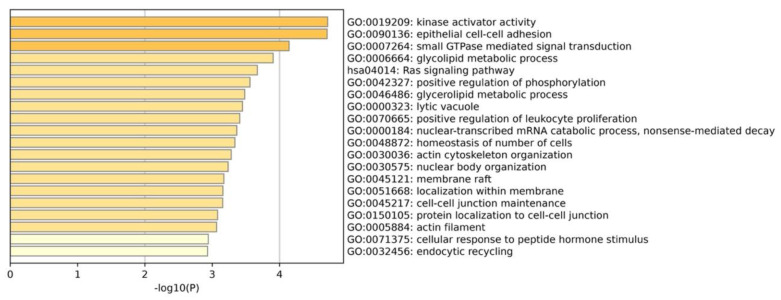
Bar graph of 20 functional enrichment categories analyzed using Metascape.

**Figure 4 animals-13-00361-f004:**
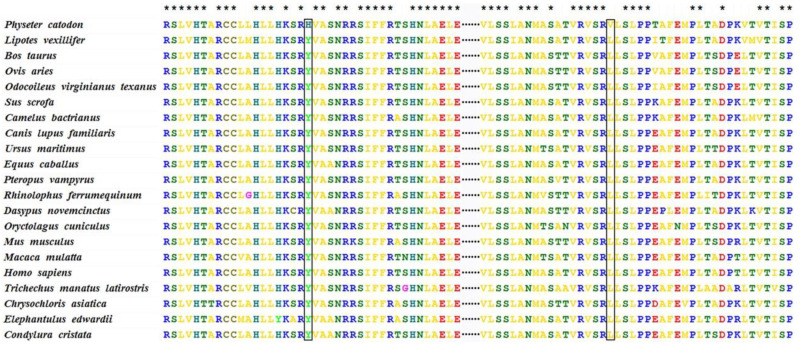
Two sperm-whale-specific missense mutations found in LIPE. The missense mutations are marked within rectangles, and the asterisk means that all species have the same amino acid type at this position.

**Figure 5 animals-13-00361-f005:**
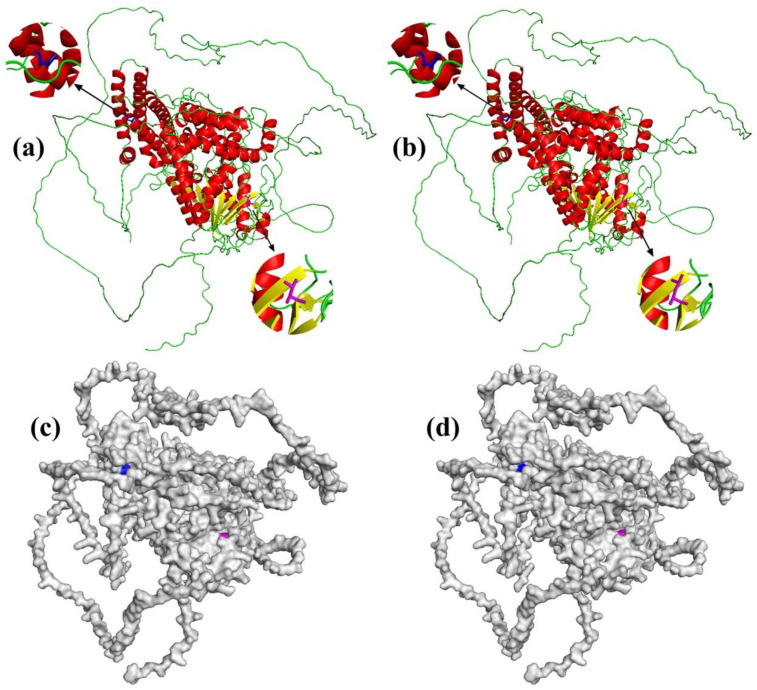
Visualization of Tyr393His and Leu567Val in LIPE: (**a**) cartoon representation of human; (**b**) cartoon representation of sperm whale; (**c**) surface representation of human; (**d**) surface representation of sperm whale.

**Figure 6 animals-13-00361-f006:**
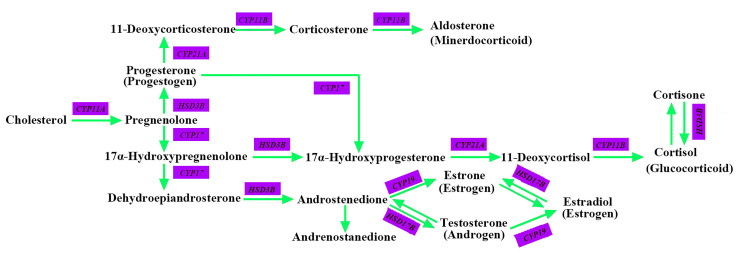
Overview of the steroid hormone biosynthetic pathway.

**Figure 7 animals-13-00361-f007:**
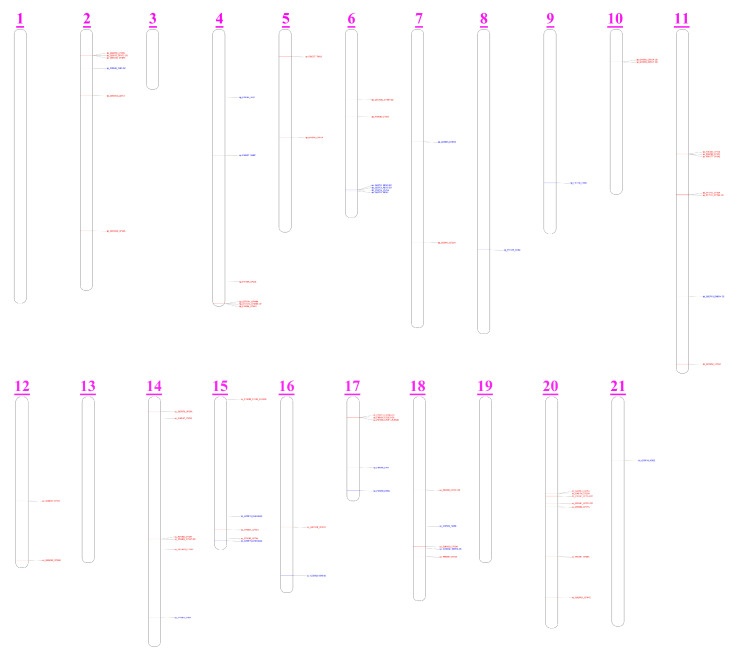
Chromosomal distribution of the functional HSDs and CYPs of the sperm whale.

**Figure 8 animals-13-00361-f008:**
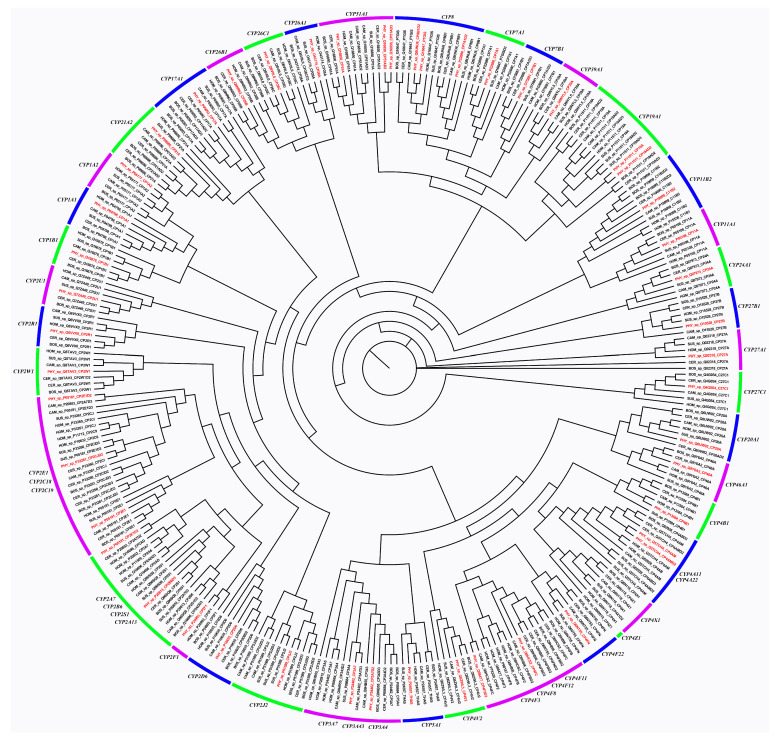
Phylogeny of the cytochrome P450 gene family of sperm whales and five other mammals. In total, 18 CYP families encompassed 52 putative genes from all six mammals, and there were 50 putative genes in 18 CYP families in sperm whales.

**Table 1 animals-13-00361-t001:** PSGs in KEGG pathways related to ambergris production.

KEGG Pathway	Map ID	Number of PSGs	Positively Selected Genes
Steroid biosynthesis	map00100	1	*LSS*
Aldosterone synthesis and secretion	map04925	4	*LIPE*
			*ADCY2*
			*ATF6B*
			*CACNA1I*
Aldosterone-regulated sodium reabsorption	map04960	1	*SLC9A3R2*

**Table 2 animals-13-00361-t002:** Sperm-whale-specific missense mutations in LIPE.

Gene	Mutation Sites (Human)	Amino Acids (Human/Sperm Whale)	Polarity	PolyPhen-2	
				HumDiv	HumVar
*LIPE*	393	Y(Tyr)/H(His)	polar/polar	0.014 (benign)	0.028 (benign)
	567	L(Leu)/V(Val)	unpolar/unpolar	0.999 (probably damaging)	0.994 (probably damaging)

## Data Availability

All the genome data of animals were downloaded from NCBI (https://www.ncbi.nlm.nih.gov/, accessed on 14 March 2022) (the accession numbers were sperm whale, GCF_002837175.2; mouse, GCF_000001635.27; common bottlenose dolphin, GCF_011762595.1; human, GCF_000001405.40; dog, GCF_014441545.1; horse, GCF_002863925.1; greater horseshoe bat, GCF_004115265.1; Arabian camel, GCF_000803125.2; pig, GCF_000003025.6; central Asian red deer, GCA_010411085.1; cattle, GCA_021234555.1; African bush elephant, GCF_000001905.1).

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
