# Peer review of "Comparative Analyses Reveal the Genetic Mechanism of Ambergris Production in the Sperm Whale Based on the Chromosome-Level Genome"

_animals, 2023, doi:10.3390/ani13030361_

Round 1

Reviewer 1 Report

The paper aims to investigate a unique and important question –– what is the genetic mechanism underlying ambergris production in the sperm whale? They use 11 publicly available genomes to identify gene family expansion/contraction and proteins under positive selection, and to perform functional enrichment and protein–protein interaction tests. They provide an in-depth discussion on the related hormone functional pathways (fecal theory), as well as the microbiota and immune network (pathology theory).

General comments:

1.     The selected 11 species for genome comparison across a very wide range of mammals, and may be too diverse to compare for the purpose of identifying the genes involved in ambergris production. e.g. Genes under positive selection in the sperm whale but not in the other 10 species may be some genes involved in land to marine adaptation, or are whale lineage-specific. If ambergris production is a species-specific trait only found in the Sperm whale, would it be more effective to compare among closely related whale species and identify the genes under selection specifically in the Sperm whale? i.e. compare and contrast the whale species that can produce ambergris and the other whale species that don’t produce ambergris.

2.     The statistical analysis of functional enrichment analysis, and protein–protein interactions enrichment analysis is a bit vague. Are multiple comparison corrections being performed? Given how much of the results and discussion lean on these significant terms, the issue of multiple comparison corrections should be addressed.

Specific comments:

Line 147: ‘The terms with the best p-values’, what are these p-values? Are they significant after multiple test correction? What are the ’20 clusters’, which are not mentioned before?

Line 149-151: “For each obtained network, a list of the top three MCODE terms with the lowest p-values was generated and assigned a unique colour.” Is this ‘color’ used in a figure, which is not cited here?

Line 194-195: Should SC5D be annotated in steroid biosynthesis or cholesterol biosynthesis (as mentioned in line 325)?

Line 218: The paper reports the ‘most enriched terms’ from the GO enrichment analysis. However, it is not indicated what p-value cutoff was used, and if those ‘top terms’ are significant after multiple test correction.

Figures and Tables:

Font size in some figures is too small: figure 1(a), y axis; figure3(b) and (c) legends; figure7 mapped genes.

Table1: If LIPE, ADCY2, ATF6B, CACNA1I are the 4 PSG in Aldosterone synthesis and secretion pathway, it is better to keep them apart from LSS (e.g. a separate line of the PSGs from each pathway).

Author Response

Reviewer 1

General comments

  1. The selected 11 species for genome comparison across a very wide range of mammals, and may be too diverse to compare for the purpose of identifying the genes involved in ambergris production. e.g. Genes under positive selection in the sperm whale but not in the other 10 species may be some genes involved in land to marine adaptation, or are whale lineage-specific. If ambergris production is a species-specific trait only found in the Sperm whale, would it be more effective to compare among closely related whale species and identify the genes under selection specifically in the Sperm whale? i.e. compare and contrast the whale species that can produce ambergris and the other whale species that don’t produce ambergris.

Answer: Thank you for your advice. We added common bottlenose dolphin (GCF_011762595.1) to the comparative genomics analysis.

  1. The statistical analysis of functional enrichment analysis, and protein–protein interactions enrichment analysis is a bit vague. Are multiple comparison corrections being performed? Given how much of the results and discussion lean on these significant terms, the issue of multiple comparison corrections should be addressed.

Answer: Thank you for your advice. Functional enrichment analysis was conducted using Metascape with default setting.

Zhou, Y.; Zhou, B.; Pache, L.; Chang, M.; Khodabakhshi, A.H.; Tanaseichuk, O.; Benner, C.; Chanda, S.K. Metascape Provides a Biologist-Oriented Resource for the Analysis of Systems-Level Datasets. Nature Communications 2019, 10, doi:10.1038/s41467-019-09234-6.

Protein–protein interactions enrichment analysis is a bit vague, so we have removed this part.

Specific comments:

Line 147:‘The terms with the best p-values’, what are these p-values? Are they significant after multiple test correction? What are the ’20 clusters’, which are not mentioned before?    

Answer: Thank you for your advice. We got the resulted of this part based on preforming identified 321 PSGs in the Metascape (http:// metascape.org). According to the methodology illustration in the paper of Metascape, the p-values were obtained by utilizing the well-adopted hypergeometric test and Benjamini-Hochberg p-value correction algorithm to identify all ontology terms that contain a statistically greater number of genes in common with an input list than expected by chance, (Zhou et al., 2019).

Zhou, Y.; Zhou, B.; Pache, L.; Chang, M.; Khodabakhshi, A.H.; Tanaseichuk, O.; Benner, C.; Chanda, S.K. Metascape Provides a Biologist-Oriented Resource for the Analysis of Systems-Level Datasets. Nature Communications 2019, 10, doi:10.1038/s41467-019-09234-6.

’20 clusters’ means 20 terms with lowest p-value. All the 20 terms and their corresponding p-values showed as colored bar were in figure 3. To lessen the ambiguousness of the Line 147, I’ve changed this sentence to ’20 enriched terms with lowest p-value were selected and exhibited’.

Line 149-151: “For each obtained network, a list of the top three MCODE terms with the lowest p-values was generated and assigned a unique colour.” Is this ‘color’ used in a figure, which is not cited here?    

Answer: Thank you for your advice. We have removed this part.

Line 194-195: Should SC5D be annotated in steroid biosynthesis or cholesterol biosynthesis (as mentioned in line 325)?    

Answer:Thank you for your advice. In our study, SC5D was not enriched in steroid biosynthesis pathway (map00100).

Line 218: The paper reports the ‘most enriched terms’ from the GO enrichment analysis. However, it is not indicated what p-value cutoff was used, and if those ‘top terms’ are significant after multiple test correction.

Answer:Thank you for your advice. This question echoes to the question in Line 147.

Reviewer 2 Report

This manuscript performed a genomic analysis in sperm whale and other ten mammalian genomes which helps improve the understandings about the genetic mechanism of ambergris production.
The authors provided a comprehensive background introduction, which is very helpful for readers to understand the state-of-the-art theory in the field. The functional enrichment and PPI network analyses of sperm whale proteins support the conclusion that ambergris production is related to steroid, terpenoid and aldosterone.
The study chose 11 mammalian genomes for comparative genomic analysis without justifying the reasons of choices. Are these organisms randomly selected? If not, what makes the authors choosing them to be compared with sperm whale? How these organisms are related to the study of ambergris production?
The authors constructed a phylogenetic tree by concatenation using maximum likelihood. This method is usually not as accurate as estimating statistically consistent species trees from gene trees while accounting for gene tree discordance using methods such as ASTRAL (https://github.com/smirarab/ASTRAL). I suggest the authors to compare these two methods and report a more accurate species tree from orthologs.
In a few places throughout the manuscript, the authors mentioned five species or six species. It sounds like that the 11 species were divided into two groups? It is not clear how or why to separate them into two groups. There is also no information to specify what organisms were in the five or six species.
Figures 3b, 3c, and 7 are not legible. Higher resolutions and legible font sizes are required.
Some figures, such as Figure 5 and Figure 7, it is not clear what the authors are trying to draw as conclusions. For example in Figure 5, visual differences are not shown between subfigures a vs b or c vs d. What was the purpose of this comparison?
There are LOTS of grammar mistakes throughout the manuscript, especially in section 4. I marked a few of them in the attached pdf. Please proofread it very carefully before submission.

Author Response

This manuscript performed a genomic analysis in sperm whale and other ten mammalian genomes which helps improve the understandings about the genetic mechanism of ambergris production. The authors provided a comprehensive background introduction, which is very helpful for readers to understand the state-of-the-art theory in the field. The functional enrichment and PPI network analyses of sperm whale proteins support the conclusion that ambergris production is related to steroid, terpenoid and aldosterone. The study chose 11 mammalian genomes for comparative genomic analysis without justifying the reasons of choices. Are these organisms randomly selected? If not, what makes the authors choosing them to be compared with sperm whale? How these organisms are related to the study of ambergris production? The authors constructed a phylogenetic tree by concatenation using maximum likelihood. This method is usually not as accurate as estimating statistically consistent species trees from gene trees while accounting for gene tree discordance using methods such as ASTRAL (https://github.com/smirarab/ASTRAL). I suggest the authors to compare these two methods and report a more accurate species tree from orthologs. In a few places throughout the manuscript, the authors mentioned five species or six species. It sounds like that the 11 species were divided into two groups? It is not clear how or why to separate them into two groups. There is also no information to specify what organisms were in the five or six species. Figures 3b, 3c, and 7 are not legible. Higher resolutions and legible font sizes are required. Some figures, such as Figure 5 and Figure 7, it is not clear what the authors are trying to draw as conclusions. For example in Figure 5, visual differences are not shown between subfigures a vs b or c vs d. What was the purpose of this comparison? There are LOTS of grammar mistakes throughout the manuscript, especially in section 4. I marked a few of them in the attached pdf. Please proofread it very carefully before submission.

Answer: Thank you for your advice. We selected 12 species for genome comparison across a very wide range of mammals and compared for the purpose of identifying the genes involved in ambergris production of the sperm whale. It would be more effective to compare among closely related whale species and identify the genes under selection specifically in the sperm whale. We added common bottlenose dolphin (GCF_011762595.1) to the comparative genomics analysis. RAxML was widely used in constructing a phylogenetic tree, and the method was employed by a lot of studies. The series relationships within Artiodactyla were recovered as (Camelidae + (Suidae + ((Delphinidae + Physeteridae) + (Cervinae + Bovidae)))) (Figure 1). Differences of sticks and surfaces can be seen in Figure 5. We have corrected the grammar mistakes. Thank you.
